# On the Numerical Treatment of the Temporal Discontinuity Arising from a Time-Varying Point Mass Attachment on a Waveguide

**George D. Manolis** * and **Georgios I. Dadoulis**

Laboratory for Experimental Strength of Materials and Structures, School of Civil Engineering,
Aristotle University of Thessaloniki, GR-54124 Thessaloniki, Greece
* Correspondence: gdm@civil.auth.gr; Tel.: +30-23-1099-5663

**Abstract:** A vibrating pylon, modeled as a waveguide, with an attached point mass that is time-varying poses a numerically challenging problem regarding the most efficient way for eigenvalue extraction. The reason is three-fold, starting with a heavy mass attachment that modifies the original eigenvalue problem for the stand-alone pylon, plus the fact that the point attachment results in a Dirac delta function in the mixed-type boundary conditions, and finally the eigenvalue problem becomes time-dependent and must be solved for a sequence of time steps until the time interval of interests is covered. An additional complication is that the eigenvalues are now complex quantities. Following the formulation of the eigenvalue problem as a system of first-order, time-dependent matrix differential equations, two eigenvalue extraction methods are implemented and critically examined, namely the Laguerre and the QR algorithms. The aim of the analysis is to identify the most efficient technique for interpreting time signals registered at a given pylon as a means for detecting damage, a procedure which finds application in structural health monitoring of civil engineering infrastructure.

**Keywords:** eigenvalue extraction; iterative methods; structural identification; complex eigenvalues; waveguides; attached masses; dynamic response

## 1. Introduction

The presence of mass attachments in structural elements such as beams and columns is quite common in civil engineering infrastructure, as is the case with pylons supporting cables, wind turbines, antennas, lights, etc., [1]. In certain cases, these mass attachments may vary with time on a scale comparable to that of the duration of the external dynamic loads. This may occur under certain conditions, as for example if the mass attachment plays the role of a damping device [2]. Specifically, pylons are infrastructure components serving a variety of functions and can be efficiently modeled as waveguides [3]. The presence of attachments on pylons leads to the concept of secondary systems, which may serve as passive or semi-active structural control devices, a typical example being liquid column dampers [4]. Furthermore, the attached mass may in turn be connected to a nonlinear spring which acts as a vibration absorber [5]. In general, it is possible to view the pylon with an attached time-variable mass as a system comprising a primary linear oscillator and a nonlinear secondary system. Much work has been done in this field from a mechanical engineering viewpoint, where the secondary system is often viewed as a Vibro-impact nonlinear energy sink [6]. As a consequence, new methods of solving the problem must be sought, such as multiple-scale expansions, which are possible for low mass ratios of the combined primary-secondary system. For heavily attached masses that are also time-dependent, recourse must be made to alternative techniques [7]. Here, we solve for the time-dependent, heavy top mass on a beam element by developing a technique based on a method originally outlined in [8] and further improved in [9], whereby the free vibration

boundary-value problem (BVP) of a beam with point masses is reduced to an eigenvalue problem through the separation of variables, but the characteristic equation involving transcendental functions is bypassed.

Instead, the solution is expanded in series involving the eigenfunctions of the beam without any attachments, plus a new set of generalized coordinates which can be recovered from the solution of single-degree-of-freedom (SDOF) equations with generalized forces at the right-hand side (RHS) that correspond to the point masses. An additional complication arises when ground vibrations are considered because the motion of an attached mass on the pylon cannot be uncoupled by modal analysis, since the absolute accelerations experienced by the mass require contributions from all modes at the same instant [10]. This necessitates a second-stage modal analysis, and if the attached mass is time-dependent, then the new eigen properties of the combined system are time-dependent since they are influenced by the mass rate of change [7]. Based on this solution methodology, the transient pylon vibrations can be Fourier transformed to extract the time dependence of the eigenfrequencies of the combined pylon-mass system.

The usual model for pylons is based on the Bernoulli–Euler fourth-order differential equation for bending, or second order differential equations for axial and torsional responses [3]. For an effective modelling of the lumped mass, it is necessary to introduce generalized functions [11], also known as Schwartz distributions, making it possible to differentiate functions whose derivatives do not exist in the classical sense. In this work, the inertia term that develops at the top of the beam is a temporal discontinuity, and the Dirac delta function appears as a force term on the RHS of the differential equation under study. It is noted that the Dirac delta function can also be used to model the presence of a crack in a structural member, which reduces stiffness and alters the dynamic response [12].

From the above discussion, it follows that a computationally demanding problem arises with the presence of point mass attachments in flexible structures. More specifically, the numerical complication arises from the fact that the point mass attachment is theoretically singular in space and may also exhibit a temporal singularity if the mass suddenly increases or decreases with time. Furthermore, if the magnitude of the mass attachment is large as compared to the mass of the supporting pylon, then the dynamic properties of the combined structural system change compared to the original, stand-alone beam. Thus, a time-dependent eigenvalue problem arises, which must be solved in either a direct or an iterative way, for every time step until the complete duration of the vibrations is covered. An additional issue is that the equations of motion yield an eigenvalue formulation that includes a non-symmetric damping matrix, which in turn renders the eigenvalues as complex quantities. Focusing on the eigenvalue analysis of the combined pylon-mass system, the coefficients of the polynomial resulting from setting the determinant of the system matrix equal to zero are recovered using the Leverrier–Faddeev algorithm [13]. Subsequently, two eigenvalue extraction methods are examined, namely the Laguerre method, which is the complex number variant of the conventional Newton–Raphson method used for real root extraction [14,15], and the QR method [16] in conjunction with the Householder algorithm [16].

Once the eigenvalue problem has been solved and followed by synthesis of the transient response of the pylon-mass system [17], then it is possible to proceed to system identification, either as a stand-alone endeavor or within the context of structural health monitoring (SHM). Typically, this entails extracting information based on the dynamic response of the monitored structure. Standard practice [18] utilizes the free vibration regime, but here the short-term Fourier transform (STFT) of the forced vibration regime is used to trace the time evolution of dominant eigenfrequencies of a pylon with a time-variable mass attached at the top. The external forcing functions are harmonic ground motion, and two cases are considered, one where the mass decreases to a nearly zero value starting from a reference value, plus the opposite case where the additional mass increases from zero to the same reference value, which can be substantial, i.e., reaching 20% of the pylon mass. These eigenfrequencies converge to the standard values computed when the additional mass has

a fixed value when ground motion ceases. The methodology presented herein is useful in extracting as much information as possible from dynamic responses for use in the SHM of typical civil engineering infrastructure ranging from pylons to bridges and buildings [19].

## 2. Non-Autonomous Dynamic Systems

In terms of the dynamic response of structural systems, the presence of attachments that lead to passive and/or active control adds terms to either the differential operator or the boundary conditions. This situation, however, presupposes that the mass and stiffness parameters of the structural system remain time-invariant [20], i.e., the system is autonomous. If this is not the case and these parameters are time-dependent, then the structural system is labeled as non-autonomous [21]. The pylon with the attached transient mass studied here belongs to the latter group, and the target is to move the eigenfrequencies of the combined system away from the dominant frequency range of the externally applied excitations [7]. In what follows, we focus on the longitudinal vibrations of the supporting pylon, although the methodology can be extended to cover transverse vibrations as well.

### 2.1. Equation of Motion for Longitudinal Vibrations

The equation of motion and boundary conditions [3] for a cantilevered pylon of length $L$ with a time-varying, lumped mass attachment $m_L(t) = m_0 \pm \mu \cdot t$ at the top $x = L$ and undergoing longitudinal vibrations, see Figure 1, are given below as follows:

$$\rho A (\ddot{u} + \ddot{x}_g) + (m_L(\ddot{u} + \ddot{x}_g) + \dot{m}_L(\dot{u} + \dot{x}_g))\delta(x - L) = EAu''$$
$$u(0,t) = 0, EAu'(L,t) = 0 \tag{1}$$

**Figure 1.** Flexible pylon of length $L$ with an attached point mass $m_L$ at the top and subjected to ground vibrations in the longitudinal direction $x$.

In the above, $E$ is the modulus of elasticity, $A$ is the cross-section area, and $\rho$ is the material density of the pylon for a continuous mass distribution. Next, $u(x,t)$ is the displacement along the x-axis and $x_g(t)$ is the time-dependent ground motion imparted at the base, with primes ($'$) and dots ($\cdot$) respectively denoting spatial and temporal derivatives. Moreover, initial conditions are assumed to be zero. It should be noted that the boundary conditions in Equation (1) have been rendered homogeneous by absorbing the lumped mass inertia effects $m_L(\ddot{u} + \ddot{x}_g) + \dot{m}_L(\dot{u} + \dot{x}_g)$ in the equation of motion using the Dirac delta function $\delta(x)$. This however changes the eigenvalue problem of the original stand-alone pylon to a more complicated one and requires an additional investigation as to the number of terms required for convergence for a modal analysis [17]. Note that the presence of a point discontinuity at boundary $x = L$ poses a more stringent convergence requirement as compared to the case of the discontinuity appearing in the interior $0 < x < L$ of the waveguide.

### 2.2. The Dirac Delta Distribution

The definition of the Dirac delta function ($\delta$) given below is heuristic, in the sense that this is not a proper function [22], but can be defined as either a measure or as a distribution:

$$\int_{-\infty}^{\infty} \delta(x)dx = 1.0 \tag{2}$$

Listed below are the translation identity and its limiting form for weight function $w(x)$ as two stations, $\xi$ and $\eta$, coalesce:

$$\int_{-\infty}^{\infty} \delta(\xi - x)\delta(x - \eta)dx = \delta(\eta - \xi),$$
$$lim_{\xi \to \eta} \int_{-\infty}^{\infty} w(x)\delta(\xi - x)\delta(x - \eta)dx = w(\eta) \tag{3}$$

### 2.3. Steel Pylon with Attached Mass

The steel pylon shown in Figure 1 is a circular cylinder with a constant cross-section, fixed at the base and with a time-varying mass attached at the top. The latter element can be considered as a tank with a valve permitting the inflow and outflow of a liquid, e.g., water. The properties of this combined structural system are listed in Table 1. Two scenarios are considered, the first for the water tank at the top being filled up from an initial mass ratio $R = 0.02$ (nearly empty) to a mass ratio $R = 0.20$ (full) with a flow rate $\mu = +1$ tn/s. The second scenario is the reverse, starting from $R = 0.20$ and dropping to $R = 0.02$ with a flow rate $\mu = -1$ tn/s. The time scale required for each of these processes to be completed is $T = 0.85$ s. The external force is a harmonic ground motion with a displacement amplitude of $x_{go} = 1 \cdot 10^{-2} [\text{m}]$.

**Table 1.** Dimensions and properties of steel pylon used as the numerical example.

| Properties | Symbol | Value | Units |
|---|---|---|---|
| Modulus of Elasticity | $E$ | 200 | GPa |
| Mass density | $\rho$ | 7.85 | tn/m$^3$ |
| Cross-section mean radius | $r$ | 0.3375 | m |
| Cross-section thickness | $d$ | 0.0875 | m |
| Cross-section area | $A$ | 0.1855 | m$^2$ |
| Pylon clear height | $L$ | 10.0 | m |
| Pylon mass | $m_P$ | 14.56 | tn |
| Attached mass | $m_L$ | 2.91 | tn |
| Rate of attached mass change | $\mu$ | $\pm 1.0$ | tn/s |

### 2.4. Dynamic Response of the Steel Pylon

Following the transposition of the attached mass inertia terms to the equation of motion, the eigenvalue problem for the pylon is now defined at time $t = 0$, where a fixed mass $m_L = m_0$ has been placed at the top end $x = L$. The characteristic equation for the pylon modeled as a waveguide under axial vibrations with the boundary condition $m_L \ddot{u}(L, t) = -EAu'(L, t)$ is:

$$\cos(k_i L) - R(k_i L)\sin(k_i L) = 1 - R(k_i L)\tan(k_i L) = 0$$
$$i = 1, 2, \ldots, \infty \tag{4}$$

In the above, the mass ratio is defined as $R = m_0/m_P$, with $m_P = \rho AL$ the mass of the stand-alone pylon and $k_i$ is the corresponding wave number. Equation (2) can be solved numerically to recover the eigenfrequencies in two stages, starting with the bisection method to extract the roots, and followed by Newton–Raphson for better convergence [14].

Given the formulation in Equation (1), separation of variables is possible and therefore the following expansion is obtained

$$u(x, t) = \Phi_i(x)q_i(t), \; i = 1, 2, \ldots, \infty \tag{5}$$

where $\Phi_i(x)$ is the eigenfunction and $q_i(t)$ is the generalized coordinate. Note that the summation convention is implied for repeated indices. For the stand-alone, cantilevered pylon, the eigenfunctions are given [10] as $\Phi_i(x) = \sqrt{2/m_P}\sin(k_i x)$, $k_i = (2i - 1)(\pi/2L)$.

### 2.5. Eigenvalue Problem Formulation

The eigenvalue problem is first expressed in terms of the aforementioned generalized coordinates $q_i(t)$, for $i = 3$ eigenvalues, which are deemed sufficient in the analysis, as follows:

$$
\begin{aligned}
[M(t)]\{\ddot{q}(t)\} + [C]\{\dot{q}(t)\} + [K]\{q(t)\} &= \{0\} \\
[M(t)] = \begin{bmatrix} 1 + 2(R+kt) & -2(R+kt) & 2(R+kt) \\ -2(R+kt) & 1 + 2(R+kt) & -2(R+kt) \\ 2(R+kt) & -2(R+kt) & 1 + 2(R+kt) \end{bmatrix}, \\
[C] = \begin{bmatrix} 2Q & -2Q & 2Q \\ -2Q & 2Q & -2Q \\ 2Q & -2Q & 2Q \end{bmatrix}, \; K = \begin{bmatrix} \omega_1^2 & 0 & 0 \\ 0 & \omega_2^2 & 0 \\ 0 & 0 & \omega_3^2 \end{bmatrix}
\end{aligned}
\tag{6}
$$

In the above, $[M(t)]$, $[C]$, and $[K]$ are the mass matrix, a damping–type matrix and the stiffness matrix, respectively, written in a normalized form. Two mass ratios are defined here as $R = m_0/m_P$ (dimensionless) and $Q = \mu/m_P$ (a normalized mass rate). Furthermore, $\omega_i$ [rad/s] are the eigenfrequencies of the stand-alone pylon. In order to recover a closed-form solution, we proceed to recast the above second-order, matrix differential equation as a first-order differential equation of double size as

$$
\begin{aligned}
[A(t)]\{\dot{y}(t)\} + [B(t)]\{y(t)\} &= \{0\}, \; \{y(t)\}^T = \lfloor \{\dot{q}(t)\}, \{q(t)\} \rfloor, \\
[A] = \begin{bmatrix} [0] & [M(t)] \\ [M(t)] & [C] \end{bmatrix}, [B] &= \begin{bmatrix} -[M(t)] & [0] \\ [0] & [K] \end{bmatrix}
\end{aligned}
\tag{7}
$$

The more compact form for the above matrix equation is

$$
\{\dot{y}(t)\} = -\left[ A^{-1}(t) \right][B(t)]\{y(t)\} = -[D(t)]\{y(t)\}
\tag{8}
$$

At this point, we opted for a time-stepping solution of Equation (8) by defining a total time interval of interest as $T = N\Delta t$ [s], with $N$ the total number of time steps $\Delta t$. Thus, $T$ is the time it takes for the mass ratio $R$ to commence from an initial value and reach a final value. During a given time instant $t_n = t_{n-1} + \Delta t$, the system matrix $[D(t_n)]$ is assumed to remain constant. Once the number of eigenvalue/eigenfunction pairs has been decided upon (in our case, $i = 3$), the eigenvalue problem is solved starting at time $t = 0$ with $[D(0)] = -\left[ A^{-1}(0) \right][B(0)]$. This is continued for all time steps $t_n$ with system matrix $[D(t_n)]$, and initial conditions equal to the final conditions for the displacement and velocity of the immediately previous time step $t_{n-1}$.

In more detail, since system matrices $[A(t)]$, $[B(t)]$ are symmetric, a solution in the form $\{y(t)\} = \{\hat{y}\} \exp(\lambda t)$, where the carat indicates the amplitude of the kinematic variables, namely of the displacement and velocity. Upon substitution in Equation (8), the eigenvalue problem assumes its standard form as follows:

$$
([A(t)]\lambda + [B(t)])\{\hat{y}\} = \{0\}
\tag{9}
$$

Solution yields the complex-valued eigenvalues $\lambda_i$, $i = 1, 2, 3$. Specifically, the condition $|[A]\lambda + [B]| = 0$ yields

$$
\begin{bmatrix} -[M] & \lambda[M] \\ \lambda[M] & \lambda[C] + [K] \end{bmatrix} \begin{pmatrix} \lambda\{\hat{y}\} \\ \{\hat{y}\} \end{pmatrix} = \begin{bmatrix} \{0\} \\ \{0\} \end{bmatrix} \Rightarrow \begin{aligned} -[M]\,\lambda\{\hat{y}\} + \lambda[M]\{\hat{y}\} &= 0 \\ (\lambda^2[M] + \lambda[C] + [K])\{\hat{y}\} &= 0 \end{aligned}
\tag{10}
$$

Since the first of the above equations is merely an identity, we impose on the second one the determinant equal to zero as $\left| \lambda^2[M(t)] + \lambda[C] + [K] \right| = 0$. This is the equation that can now be solved for every time increment from the onset of motion at $t = 0$. The time-dependent eigenvalue code is depicted in Table 2 and all computations were carried out using the Python [23] software.

**Table 2.** Formation of system matrices $[M(t_i)]$, $[C]$, $[K]$, $[D(t_i)]$.

---

**function MassMatrix**$(t_i)$
**Input** : $t_i$
**Output** : $M(t_i)$
 $M =$ **zeros**$(N, N)$
 **for** $i = 1, \ldots, N$ **do**
  **for** $j = 1, \ldots, N$ **do**
   **if** $(i + j) \% 2 \mathbin{!} = 0$ **then**
    $M[i, j] = -2 * R - 2 * k * t_i$
   **end if**
   **if** $(i + j) \% 2 == 0$ **then**
 $M[i, j] = 2 * R + 2 * k * t_i$
   **end if**
   **if** $(i + j) \% 2 == 0$ **then**
 $M[i, j] = 1 + 2 * R + 2 * k * t_i$
   **end if**
  **end for**
 **end for**
 **return** $M$

**function** : **DampingMatrix**
**Output** : $C$
 $C =$ **zeros**$(N, N)$
 **for** $i = 0, \ldots, N$ **do**
  **for** $j = 0, \ldots, N$ **do**
   **if** $(i + j) \% 2 \mathbin{!} = 0$ **then**
    $C[i, j] = -2 * k$
   **end if**
   **if** $(i + j) \% 2 == 0$ **then**
    $C[i, j] = 2 * k$
   **end if**
  **end for**
 **end for**
 **return** $C$

**function** : **StiffnessMatrix**
**Output** : $K$
 $K =$ **zeros**$(N, N)$
 **for** $i = 1, \ldots, N$ **do**
  **for** $j = 1, \ldots, N$ **do**
   **if** $i == j$ **then**
    $K[i, j] = i * \frac{\pi}{L} * \sqrt{\frac{E}{\rho}}$
   **end if**
  **end for**
 **end for**
 **return** $K$

---

Assembly and solution using the $2N \times 2N$ matrix $[D(t_i)]$ for a given time step, see Equation (9):

$$[D(t_i)] = -\left[A^{-1}(t_i)\right][B(t_i)] = \begin{bmatrix} -\left[M^{-1}(t_i)\right][C] & -[M^{-1}(t_i)][K] \\ [I] & [0] \end{bmatrix}$$

```
function : DeltaMatrix
Input : tᵢ
Output: D (tᵢ)
 D = zeros(2N, 2N)
 D₁ = −inverse(MassMatrix(tᵢ)) ∗ DampingMatrix
 D₂ = −inverse(MassMatrix(tᵢ)) ∗ StiffnessMatrix
 D₃ =  identity(N, N)

 for i = 0, . . . , N  do
   for j = 0, . . . , N  do
     D[i, j] = D₁[i, j]
   end for
   for j = N, . . . , 2N  do
     D[i, j] = D₂[i, j − N]
   end for
 end for

 for i = N, . . . , 2N  do
   for j = 0, . . . , N  do
     D[i, j] = D₃[i − N, j]
   end for
 end for
 return  D
```

## 3. Eigenvalue Extraction for Non-Autonomous Dynamic Systems

The general form of the eigenvalue problem is shown below, from which it is possible to recover the eigenvalues as the roots of the polynomial $p(\lambda)$ that results from setting the determinant of the matrix system equal to zero:

$$p(\lambda) = \det([D] - \lambda[I]) = 0 \tag{11}$$

The first step in the solution of the characteristic equation which emerges from Equation (11) requires the computation of the coefficients of the resulting polynomial. This is achieved by using the Leverrier–Faddeev algorithm [24], which produces a sequence of successive matrices replacing $[D]$ until it becomes diagonal and then computes its trace as the sum of the diagonal terms. The trace is then used as input for the Laguerre algorithm [25] and the output is $N$ pairs of complex conjugate roots, where $N$ is the number of eigenmodes retained in the representation of the dependent variable, see Equation (5):

$$\lambda_i = \xi_i \omega_i \pm j\omega_i \sqrt{1 - \xi_i^2} = \delta_i \pm j\omega_{d,i} \quad j = \sqrt{-1}, \quad i = 1, 2, 3, \ldots \tag{12}$$

In the above, $\omega_{d,n}$ (rad/s) are the damped eigenfrequencies of the stand-alone pylon for Rayleigh damping, with $\xi_i$ the viscous damping ratio. In order to recover real-valued eigenfrequencies (in $Hz$) from Equation (12), the amplitude of $\lambda_i$ is divided by $2\pi$, resulting in

$$f_i = \sqrt{(\xi_i \omega_i)^2 + \omega_i^2 (1 - \xi_i^2)} / 2\pi, \quad i = 1, 2, 3, \ldots \tag{13}$$

### 3.1. The Laguerre Method

Laguerre's method [25] expresses a polynomial of order $n$ in a recursive form as

$$p_0(x) = a_n, \ p_i(x) = a_{n-i} + x p_{i-1}(x), \ i = 0, 1, \ldots, n-1, \ n = 2N \tag{14}$$

The derivatives of the polynomial are computed as

$$p_i'(x) = p_{i-1}(x) + x p_{i-1}'(x), \ p_i''(x) = 2p_{i-1}'(x) + x p_{i-1}''(x) \tag{15}$$

Thus, bringing the polynomial in the form

$$p_n(x) = (x - r)(x - q)^{n-1} \tag{16}$$

it becomes obvious that there is a root at $x = r$ and there are $n - 1$ roots at $x = q$. The next step is to take the derivative of Equation (16) as

$$\begin{aligned} p_n'(x) &= (x - q)^{n-1} + (n - 1)(x - r)(x - q)^{n-2} \Rightarrow \\ G(x) &= p_n'(x)/p_n(x) = 1/(x - r) + (n - 1)/(x - q) \end{aligned} \tag{17}$$

Following that, one proceeds with the second derivative:

$$p_n''(x)/p_n(x) - \left(p_n'(x)/p_n(x)\right)^2 = -1/(x - r)^2 - (n - 1)/(x - q)^2 \tag{18}$$

By defining $H(x) = G^2(x) - p_n''(x)/p_n(x)$ and manipulating terms finally yields the recursive Laguerre formula for root $r$ is as follows:

$$x - r = n / \left\{ x \pm \sqrt{(n - 1)(nH(x) - G^2(x))} \right\} \tag{19}$$

The improved root $r$ determined by choosing the sign results in the larger magnitude of the denominator.

### 3.2. The Leverrier–Faddeev Method

For the purpose of numerical implementation, Table 3 below gives the structure of the Leverrier–Faddeev algorithm, which accepts as input matrix $[D]$ and returns as output coefficients $\{a\} = [a_n, a_{n-1}, \ldots, \alpha_2, \alpha_1, 1]$ of the characteristic polynomial $p(\lambda) = \lambda^n + a_1 \lambda^{n-1} + \ldots + a_{n-1}\lambda + a_n$, where $n = 2N$ and $a[0] = a_n$, $a[1] = a_{n-1}$, etc.

**Table 3.** The Leverrier–Faddeev method.

| |
| --- |
| **function** : **Leverrier_Faddeev** |
| **Input** : $D(t_i)$ |
| **Output** : $a$ |
| $a = \mathbf{zeros}(2N + 1)$ |
| $D_1 = D$ |
| $a = \mathbf{insert}(2N, 1)$ |
| $a = \mathbf{insert}(2N - 1, -\mathbf{trace}(D_1))$ |
| **for** $i = 2, \ldots, 2N$ **do** |
| $\quad \Delta_i = \Delta \cdot (\Delta_{i-1} + a[2N - i + 1] \cdot I)$ |
| $\quad a = \mathbf{insert}\left(2N - i, -\frac{\mathbf{trace}(\Delta_i)}{i}\right)$ |
| **end for** |
| **return** $a$ |

Before sending these coefficients as input to the Laguerre algorithm for computing the roots of the characteristic polynomial, an ancillary function that computes its derivatives is necessary. This function is labelled *EvalPoly* and appears in Table 4 below, while Table 5 gives the subroutine used for the polynomial root recovery.

Note that possible roots $\{x\}$ at time zero are actually the vector of the wavenumbers of the pylon:

$$x_n = \pm\big((2n + 1)(\pi/2L)c_p\big)j, \quad n = 0, 1, \ldots, N, \quad j = \sqrt{-1} \tag{20}$$

where $c_P = \sqrt{E/\rho}$ is the longitudinal wave velocity.

The disadvantage of the Laguerre algorithm is that small changes in the coefficients $\{a\} = [a_n, a_{n-1}, \cdots, \alpha_2, \alpha_1, 1]$ of the characteristic polynomial $p(\lambda) = \lambda^n + a_1\lambda^{n-1} + \cdots + a_{n-1}\lambda + a_n$ may lead to large variations about the correct values of the roots. Figure 2 shows

how these coefficients change as the eigenvalue problem for the pylon with the attached mass is solved every time step $t_i = t_i + \Delta t$, by retaining $N = 2$ eigenvalues. This leads to a $4 \times 4$ system of equations, which is the minimum size for reliable root recovery. More specifically, the left column in Figure 2 is for the decreasing mass rate and the right column is for the increasing mass rate.

**Table 4.** Algorithm EvalPoly for polynomial derivatives.

---

**function** : **EvalPoly**
**Input** : $a, x$ :$a$ **are the coefficients of the** $p(\lambda)$, $x$ **is an arbitrary number**
**Output: p, dp, ddp**
  $n = \mathbf{len}(a) - 1$
  $p = a[n]$
  $dp = 0.0 + 0.0\,\mathbf{j}$
  $ddp = 0.0 + 0.0\,\mathbf{j}$
  **for** $i = 1, \ldots, n + 1$ **do**
    $ddp = ddp \cdot x + 2 \cdot dp$
    $dp = dp \cdot x + p$
    $p = p \cdot x + a[n - i]$
  **end for**
  **return**   $p, dp, ddp$

---

**Table 5.** Polynomial roots by Laguerre's method.

---

**function** : **Laguerre**
**Input** :  $a, x, tol$ :$a$ **are the coefficients of the** $p(\lambda)$, $x$ **are possible roots, tol is the tolerance**
**Output** : $x, iteration$
  $n = \mathbf{len}(a) - 1$
  $iteration = 1$
  **for** $iteration = 1, \ldots, 30 \ do$
    $p, dp, ddp = \mathbf{EvalPoly}(a, x)$
    **if** $|p| < tol$ **then**
      **return** $x, iteration$
    **end if**

    $G = \dfrac{dp}{p}, \quad H = G^2 - \dfrac{ddp}{p}, \quad F = \sqrt{(n - 1)\left[nH - G^2\right]}$

    **if** $|G + F| > |G - F|$ **then**
      $dx = n/(G + F)$
    **else**
      $dx = n/(G - F)$
    **end if**
    $x = x - dx$
    **if** $|dx| < tol$ **then**
   **return**   $x, iteration$
    **end if**
  **end for**
  $\mathbf{print}(\mathbf{'Too\ many\ iterations'})$

---

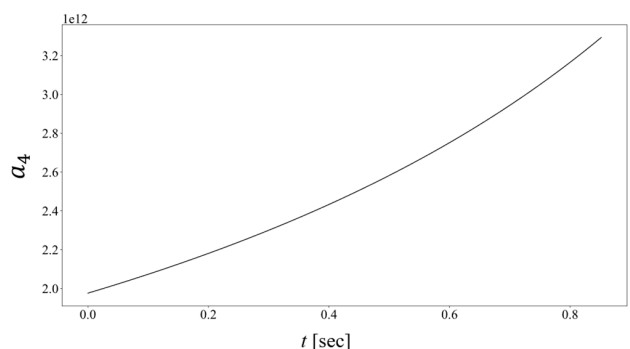 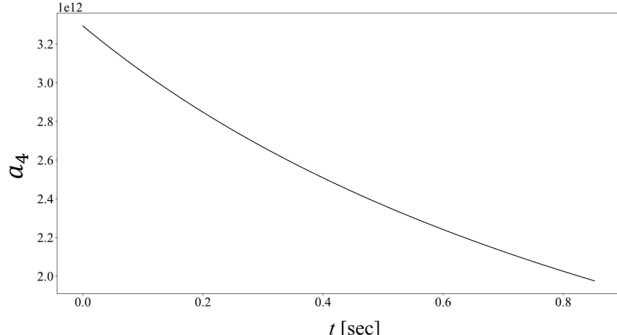

**Figure 2.** *Cont.*

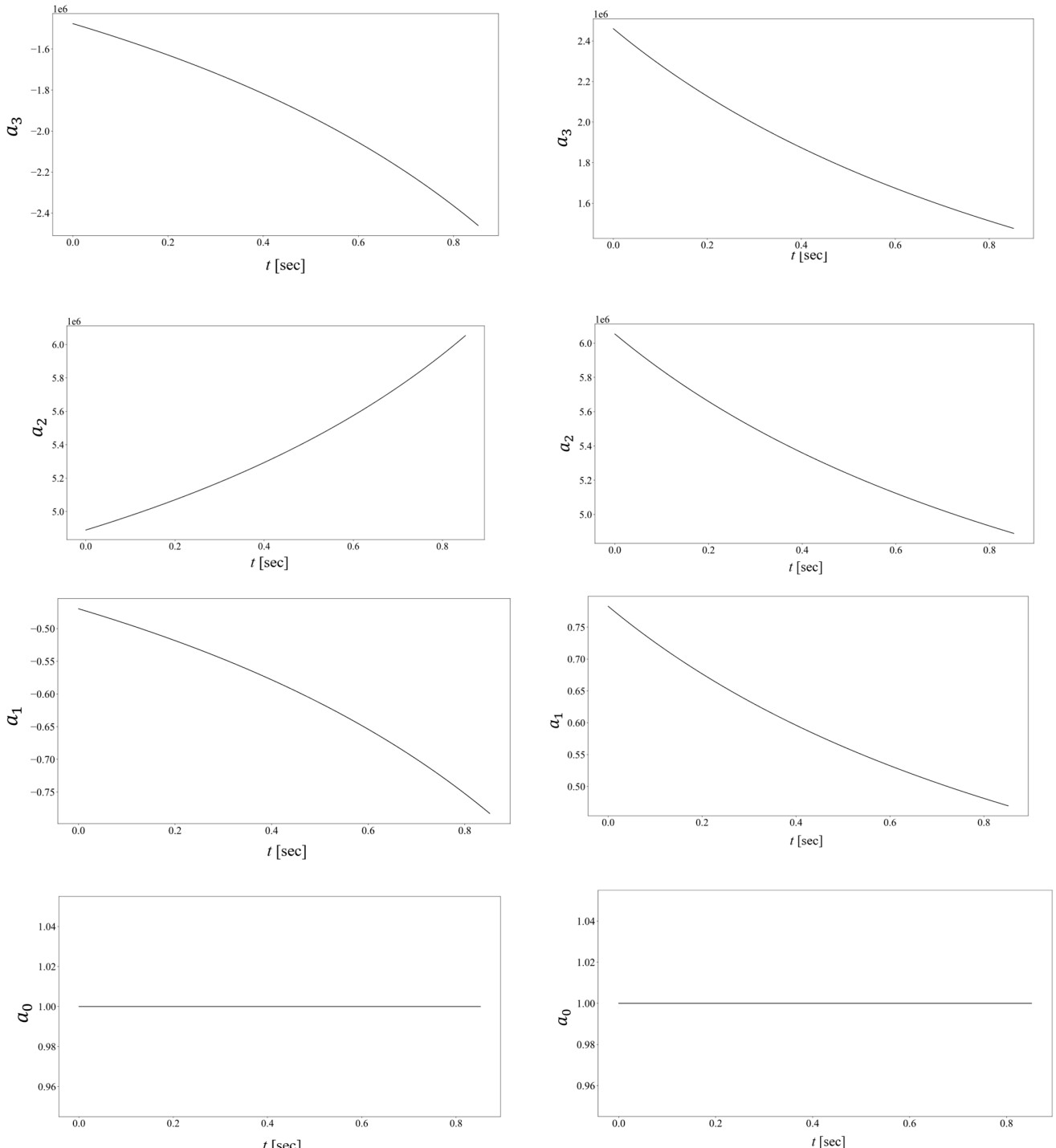

**Figure 2.** Time evolution of the characteristic polynomial coefficients $a_n$: Left column is the attached mass decrease from $R = 0.20$ to $R = 0.02$; right column is the attached mass increase from $R = 0.02$ to $R = 0.20$.

### 3.3. The QR Method

The QR method [26,27] transforms a square matrix $[D]$ into the product of an orthogonal matrix $[Q]$ (for which its transpose is the same as its inverse, i.e., $[Q^T] = [Q^{-1}]$) times an upper triangular matrix $[R]$, such that $[D] = [Q][R]$. In order to compute the eigenvalues

of $[D] \in \mathrm{R}^{n \times n}$, one begins by setting an initial matrix $[D_0] = [D]$ and performing the following iteration cycle:

$$
\begin{aligned}
[D_0] &= [Q_1][R_1] \;\rightarrow\; [D_1] = [R_1][Q_1] \\
[D_1] &= [Q_2][R_2] \;\rightarrow\; [D_2] = [R_2][Q_2] \\
&\qquad\qquad \vdots \\
[D_{n-1}] &= [Q_n][R_n] \;\rightarrow\; [D_n] = [R_n][Q_n]
\end{aligned}
\tag{21}
$$

After a number of iterations, matrix $[D_n]$ converges to a lower triangular form:

$$
[D_n] =
\begin{bmatrix}
\mathrm{X} & \mathrm{X} & * & * & * & * & * & * \\
\mathrm{X} & \mathrm{X} & * & * & * & * & * & * \\
0 & 0 & \mathrm{X} & \mathrm{X} & * & * & * & * \\
0 & 0 & \mathrm{X} & \mathrm{X} & * & * & * & * \\
0 & 0 & 0 & 0 & \mathrm{X} & \mathrm{X} & * & * \\
0 & 0 & 0 & 0 & \mathrm{X} & \mathrm{X} & * & * \\
0 & 0 & 0 & 0 & 0 & 0 & \mathrm{X} & \mathrm{X} \\
0 & 0 & 0 & 0 & 0 & 0 & \mathrm{X} & \mathrm{X}
\end{bmatrix}
\tag{22}
$$

The eigenvalues of the original matrix system are now recovered in closed form for each of the $2 \times 2$ submatrices appearing along the diagonal. These are complex conjugate pairs and are given by

$$
\lambda_m, \overline{\lambda}_m = 0.5 * \mathrm{tr} \pm \left( 0.5 \sqrt{4\mathrm{det} - \mathrm{tr}^2} \right) j, \; j = \sqrt{-1}
\tag{23}
$$

for the $m - th$ pair, where $tr$ and $det$ are the trace and determinant of the submatrix. As a convergence criterion one may define that the absolute value of both the trace and the determinant of any $2 \times 2$ submatrix coming from the last $(n - th)$ iteration $[D_n]$ is within $\varepsilon = 10^{-12}$ from the corresponding one computed from the immediately previous iteration $[D_{n-1}]$.

### 3.4. The Householder Algorithm

In general, there are three basic methods for producing the QR factorization, namely the (a) Householder, (b) Givens, and (c) Gram–Schmidt. The choice of the particular method depends on the form of the original matrix $[D]$ (e.g., sparse versus fully populated) [28]. Here we employ the Householder method, which is an algorithm requiring as input vector $\{x\}$ plus component $k$ and producing as output matrix $[H]$, so that the new vector $[H]\{x\}$ will have the remaining $n - k$ components equal to zero. Table 6 below gives the coded form of the Householder algorithm.

**Table 6.** The Householder algorithm.

```
function : Householder
Input : x, k: vector {x} and element k
Output : H
    s = √(x_k² + x_{k+1}² + ... + x_n²)
    sgn = x_k / |x_k|
    s̄ = √(2s(s + |x_k|))
    u = 0
    u_k = (1/s̄)(x_k + sgn·s)
     for j = k + 1, ..., n do
      u_j = (1/s̄) x_j
     end for
    H = I − 2uu^T
    return H
```

### 3.5. The QR-Householder Method

The flow of iterations begins by factorization using the Householder algorithm by starting as input vector $\{x\}$ the first column of matrix $[D]$ and receiving as output matrix $[H_1]$ with the $n-1$ elements of the first column of the matrix product $[H_1][D]$ equal to zero. In our case, for the $4 \times 4$ matrix $[D]$ we have

$$[H_1][D] = \begin{bmatrix} * & * & * & * \\ 0 & * & * & * \\ 0 & * & * & * \\ 0 & * & * & * \end{bmatrix} = [D_1] \tag{24}$$

The next step produces matrix $[H_2]$ with zero $n-2$ elements of the second column of $[H_2][D_1]$, followed by a third step that results in an upper triangular matrix:

$$[H_2][D_1] = \begin{bmatrix} * & * & * & * \\ 0 & * & * & * \\ 0 & 0 & * & * \\ 0 & 0 & * & * \end{bmatrix} = [D_2], \ [H_3][D_2] = \begin{bmatrix} * & * & * & * \\ 0 & * & * & * \\ 0 & 0 & * & * \\ 0 & 0 & 0 & * \end{bmatrix} = [D_3] = [R]. \tag{25}$$

Arranging terms gives $[H_3][H_2][H_1][D] = [R]$, so that $[D] = ([H_1][H_2][H_3])[R] = [Q][R]$, keeping in mind that all $[H]$ matrices are orthogonal and symmetric. The programmed structure of this method is given in Table 7 below.

**Table 7.** The QR-Householder algorithm.

```
function : QR_Householder
Input : D
Output : Q, R
  R = D
  Q = I
  for i = 0, . . . , n − 1 do
    H = Householder(D[:, i], n − i)
    R = H·R
    Q = Q·H
  end for
return Q, R
```

In order to determine the convergence rate of the solution of Equation (1) with the Dirac delta distribution formulation, both Laguerre and Householder-QR algorithms are tested as the size of the system matrix increases. Specifically, the ratio $\hat{f}_i^N / f_i$, is examined, where $\hat{f}_i^N$ is the $i-th$ eigenfrequency recovered by either algorithm for a system matrix of size $N \times N$ with $f_i$ is the exact value. Thus, the left column in Figure 3 plots the values of the first four eigenfrequencies recovered by the aforementioned algorithms (in red color), concurrently with the analytical solution (in black color) for a nearly zero mass ratio of $R = 0.02$. At the same time, the left column in Figure 3 plots the rate of convergence as the number $N$ increases. Similarly, Figure 4 plots the same information, but for a substantial mass ratio of $R = 0.20$. In both cases, the rate of convergence is satisfactory and after $N = 5$ terms, the error is negligible. Note that both algorithms give the same ratio for size $N = 2$, past which the Laguerre is no longer accurate and only the Householder-QR algorithm is used.

### 3.6. Time Evolution of the Pylon Eigenvalues

When working with $i = 5$ terms in Equation (5), ten roots $\lambda_i$ are recovered by the QR Householder method for the eigenvalues, which appear as complex conjugate pairs. These are now plotted in Figure 5 as $f_i = \omega_i / 2\pi$ [Hz] (see Equation (13)) and the real part $\delta_i$ (see Equation (12)) for every time step $\Delta t = 10^{-2}$ s starting from time zero to time $T$, past which

the flow has stopped and we either have a stand-alone pylon or a pylon with a fixed top mass $m_0$. We clearly observe in Figure 5 the evolution of the eigenvalues for longitudinal vibrations of the pylon as the mass of the water tank changes over time. As expected, this system becomes stiffer as its mass decreases (Figure 5a) and becomes more flexible as its mass increases (Figure 5b). In either case, the eigenfrequencies converge to their expected values when the system is stationary.

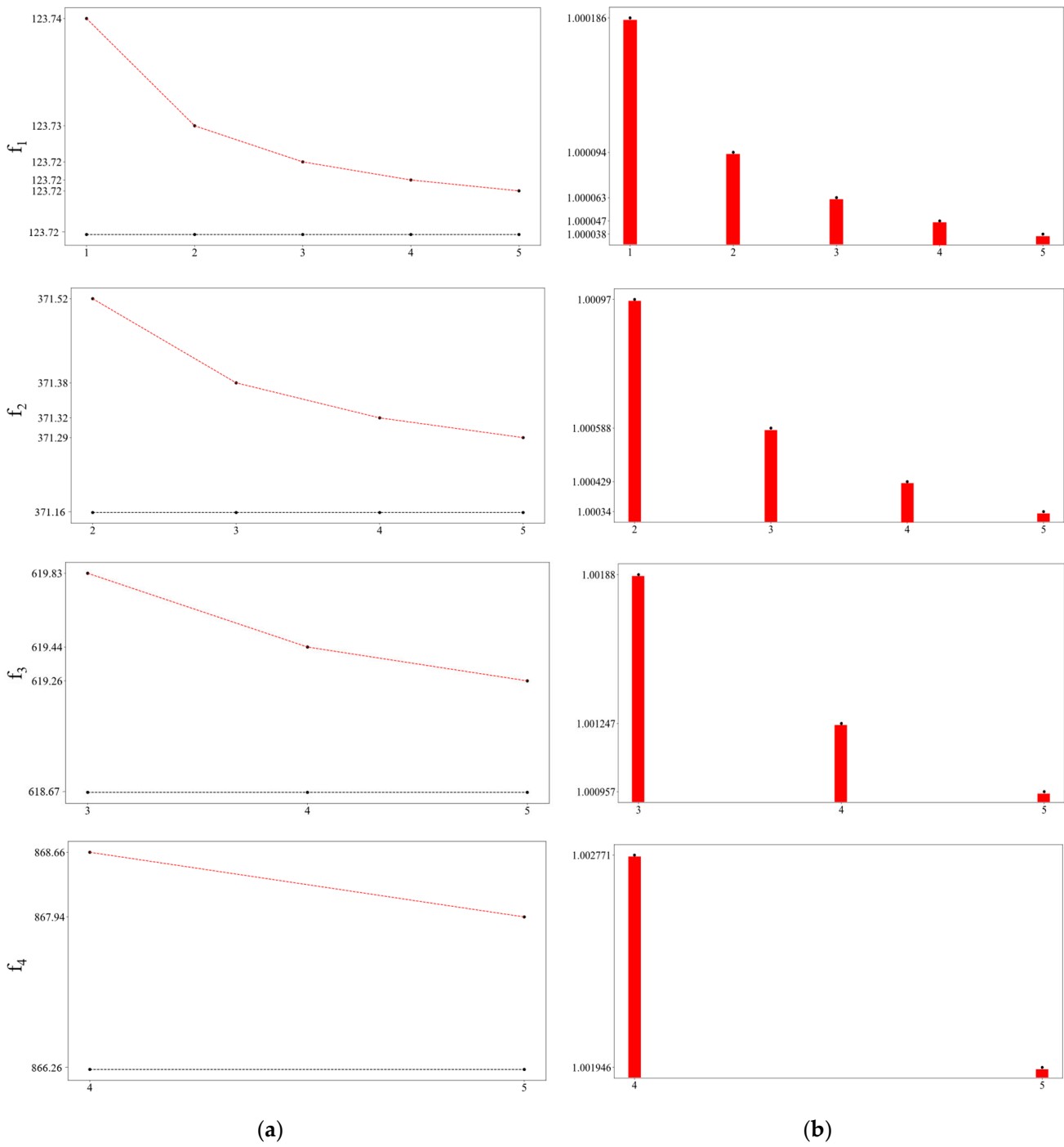

**(a)** **(b)**

**Figure 3.** The first four eigenfrequencies of the pylon with an attached transient mass for mass ratio $R = 0.02$ and at time $t = 0$: (**a**) Left column plots the eigenfrequencies as the number of iterations increases (red color) and the exact values (black color) (**b**) Right column plots the corresponding rate of convergence.

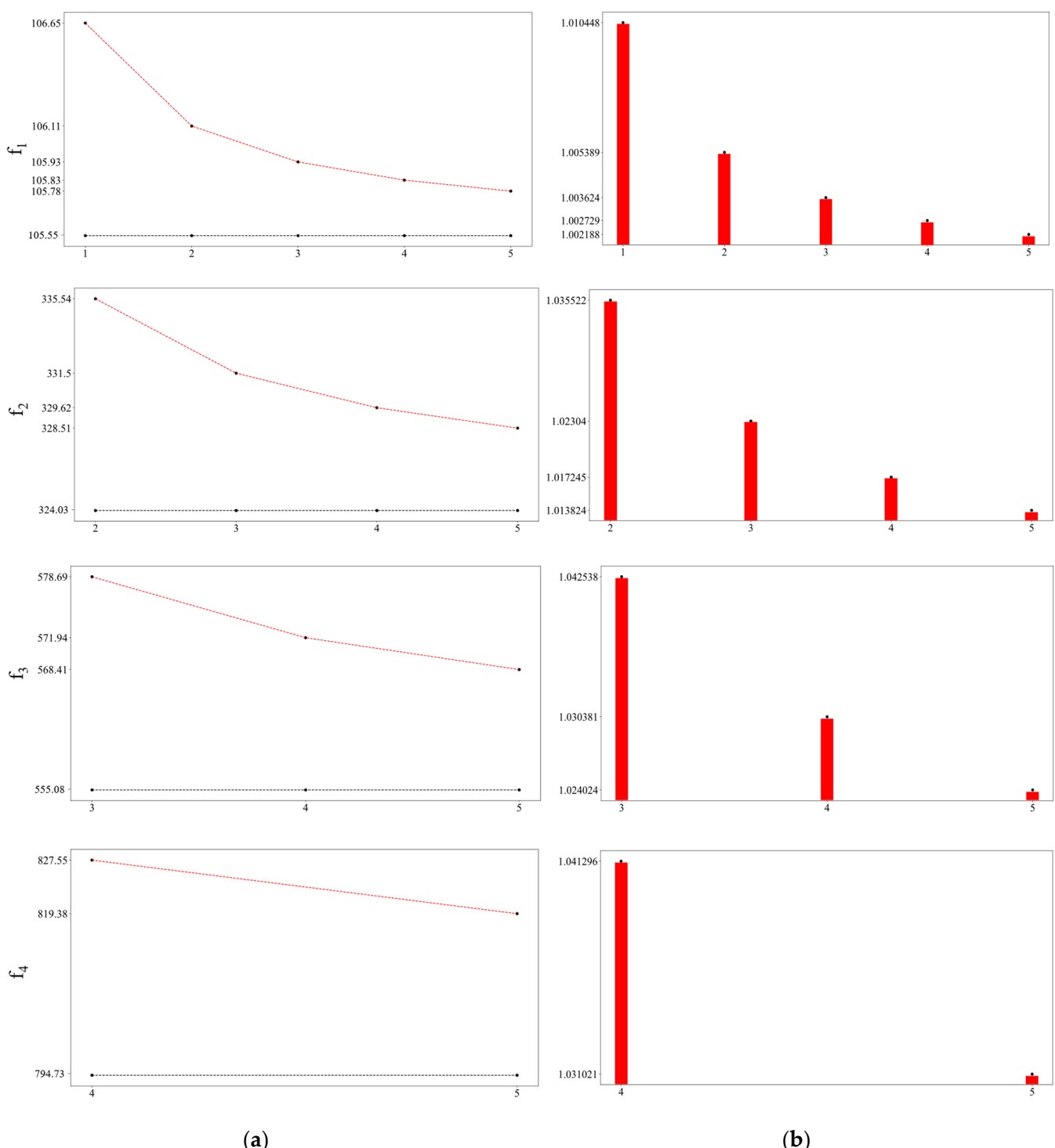

(**a**)                                           (**b**)

**Figure 4.** The first four eigenfrequencies of the pylon with an attached transient mass for mass ratio $R = 0.20$ and at time $t = 0$: (**a**) Left column plots the eigenfrequencies as the number of iterations increases (red color) and the exact values (black color); (**b**) right column plots the corresponding rate of convergence.

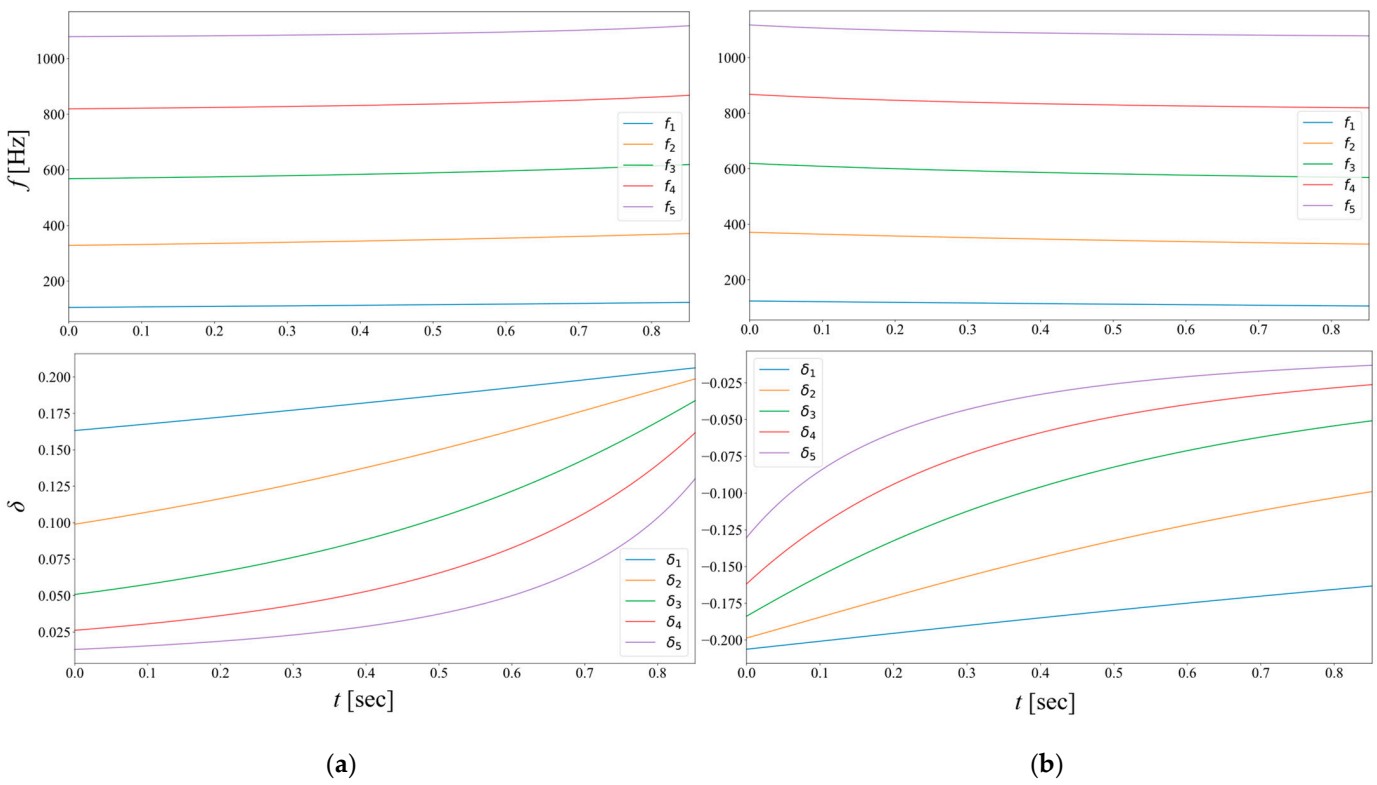

**(a)**

**(b)**

**Figure 5.** Pylon-mass system eigenfrequency $f_i$, $\delta_i$, $i = 1, 2, 3, 4, 5$ evolution with time for the case of (**a**) decreasing fluid mass ($\mu > 0$) and (**b**) increasing fluid mass ($\mu < 0$).

## 4. Derivation of Spectograms

For SHM purposes, it is necessary to trace the evolution of the eigenvalues of the pylon as its attached mass varies with time from tracing the transient displacement $u(x, t)$ at a given station $x$. In reference to the pylon described in Section 2, the dynamic characteristics of the combined pylon-attached mass system vary with time and the resulting transient response is characterized from a stochastic viewpoint as non-stationary. This requires subjecting the transient signal to a series of continuous transformations to the frequency domain by implementing the short time Fourier transformation (STFT). For sufficiently small-time steps, the signal recorded within a given time interval can be considered as stationary. To this purpose, we employ the Fourier transform [22] in the following form:

$$F_u(f, t) = \int_{-\infty}^{+\infty} u(x = L, \tau) e^{-i(2\pi f)\tau} W(\tau - t) \, d\tau \tag{26}$$

In the above, $F_u(u, t)$ is a Fourier transform of the pylon displacement at $x = L$, see Equation (5), and is known as a spectrogram. Moreover, $f = \omega/2\pi$ is the frequency (in Hz) and function $W$ is the Hanning window within a time interval $\Delta t = \tau - t$:

$$W(n) = 0.5 - 0.5 \cos(2\pi j/J), \ \ 0 \le j \le J \tag{27}$$

The width of the Hanning window is $H = J + 1$ and should cover at least one cycle of vibration. Fixing the window's width, as well as the placement of two consecutive windows, is a trial-and-error procedure for achieving optimal results.

The derivation of a spectrogram is now given for both cases where the mass attachment on the pylon is either full of a liquid that is allowed to drain, or empty and allowed to fill. At first, Figure 6 plots the transient response at the top of the pylon following a modal analysis based on the results of the eigenvalue extraction problem. Note that when alternative numerical methods such as the Runge–Kutta method are used for computing

the transient displacements, then the time evolution of the eigenvalues cannot be recovered directly, but require additional processing by various transformation techniques. The input is a time harmonic ground motion with an excitation frequency of $f = 10$ Hz. As the attached mass decreases with time, the structural system becomes stiffer and the response of the pylon slowly decreases by about 18% at the end of the time interval of about 0.85 s required to completely remove the mass, see Figure 6a. At the same time, the corresponding spectrogram, which reproduces the analytically computed time evolution of the combined system eigenfrequencies that was previously obtained, shows a 30% change as the system becomes stiffer due to the decreasing top mass. Finally, the white line at the bottom of the plot corresponds to the forcing frequency of 10 Hz. The exact opposite behavior is shown in Figure 6b, with the response increasing and the eigenfrequencies dropping as the system becomes more flexible with time.

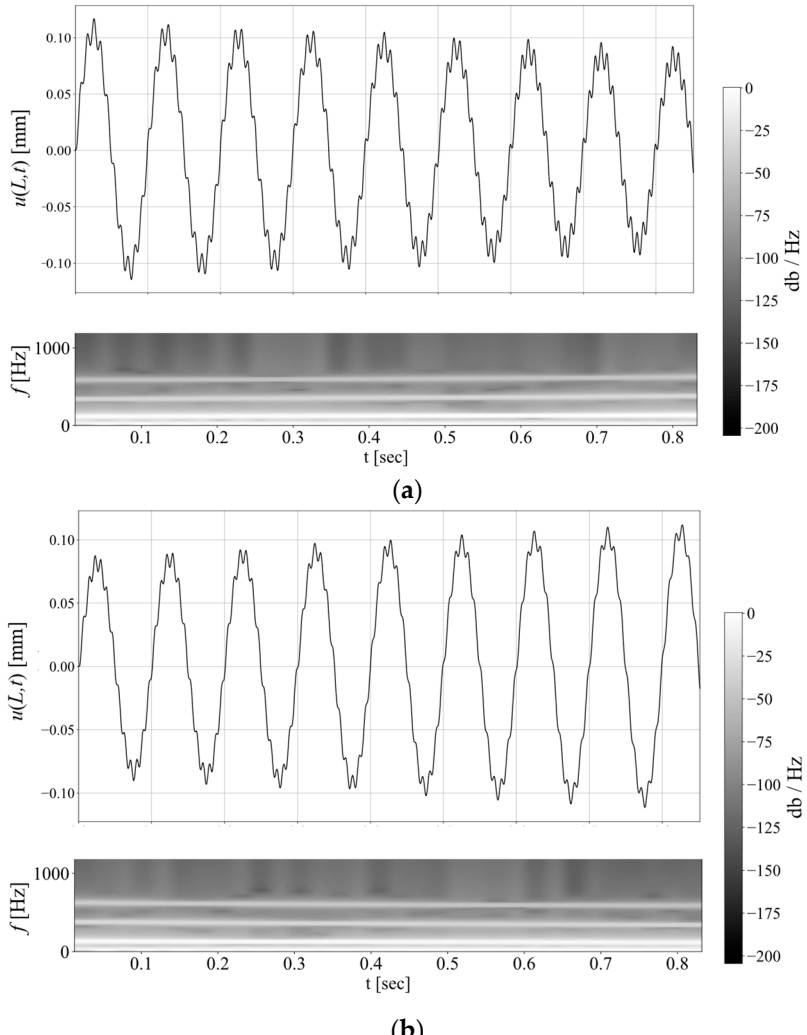

**Figure 6.** Axial displacement at the top of the pylon for harmonic ground motion $x_g(t) = x_{go} \sin(2\pi{\cdot}10t)$ and computed spectrogram depicting the time evolution of the first three pylon-mass system eigenfrequencies $f_i(t)$: (**a**) Decreasing top mass case ($\mu < 0$); (**b**) increasing top mass case ($\mu > 0$).

*Comparison of Results*

In this section, we compare the results for the eigenvalues of the pylon with a time variable mass, as computed by the iterative methods of solution, see Section 3, and also indirectly from the spectograms given above. Note that these spectograms were derived from a numerical solution of the equations of motion using the Runge–Kutta method that yielded the displacement function, to which the short-term Fourier transform was

applied. The comparison is limited to three eigenvalues, because it becomes exceedingly difficult to derive higher-order eigenvalues from the results given by the Runge–Kutta method. Figure 7 depicts the results of this comparison study, which shows good agreement between the pylon eigenvalues recovered from these two basic categories of solving transient problems.

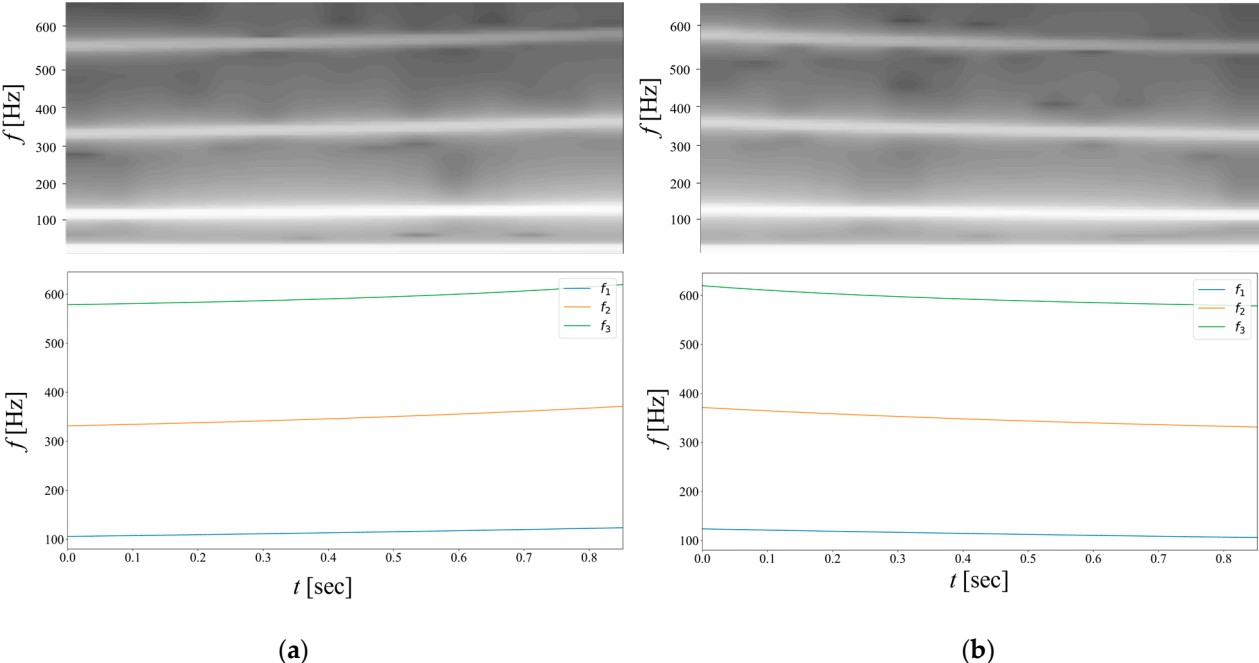

**Figure 7.** First three pylon-variable mass structural system eigenvalues: (**a**) negative mass rate $\mu = -1.0$ (tn/s) and (**b**) positive mass rate $\mu = 1.0$ (tn/s): Top graphs derive from a Runge–Kutta solution of the equations of motion; bottom graphs derive from the iterative methods of solution of the eigenvalue problem.

## 5. Discussion and Conclusions

This work investigates the applicability of iterative eigenvalue extraction algorithms for the purpose of structural identification. More specifically, the application example is a flexible, metallic cantilevered pylon with a time-dependent mass attachment. This specific attachment renders the eigenvalue problem time-dependent for the time duration of the action of the applied loads. Once the transient motion at a given station on the pylon is synthesized by modal analysis, the application of the short-time Fourier transform yields a clear picture of the manner in which the eigenvalues of the combined pylon-mass structural system vary during the action of ground-induced motions.

Focusing on the eigenvalue analysis of the combined pylon-mass system, at first the coefficients of the polynomial resulting from setting the determinant of the first-order matrix system formulation equal to zero are recovered using the Leverrier–Faddeev algorithm. This is a recursive method for calculating the coefficients of the characteristic polynomial of a square matrix [13]. However, these coefficients show large differences in their absolute values rendering any subsequent method for extracting eigenvalues potentially unstable. Therefore, the eigenvalue extraction is first performed by the Laguerre method, which is the complex number variant of the conventional Newton–Raphson method used for real root extraction [14,15]. For the Laguerre method, an additional complication is computing the correct initial values at the beginning of each time step, which are the final values of the immediately previous time step. Next, the QR method [16] is implemented in conjunction with the Householder algorithm, which is the best option in terms of both accuracy and efficiency, while all limiting cases are correctly reproduced when the attached mass attains a fixed value. In general, it is estimated that three eigenvalues are necessary

for any subsequent modal analysis when the attached point mass is at a boundary, which in our case is the top of the pylon. If the mass is attached at an intermediate station on the pylon, then two eigenvalues are sufficient, which implies one can use the Laguerre method as well.

At first, an analytical solution for the axial motion of the pylon modeled as a distributed mass system is derived, with the formulation augmented to handle an attached, time-dependent mass attachment at the top which is heavy and cannot be viewed as a secondary system. The solution is obtained by modal analysis following the eigenvalue extraction. What complicates the solution is the fact that the mass is time-dependent, i.e., may either increase or decrease during the time interval the external forces are active. This requires a second-tier solution of a time-dependent eigenvalue problem cast as a system of first order differential equations with non-constant coefficients. Next, the eigenvalues are plotted against the time duration of the increasing/decreasing fluid flow from a container at the top of the pylon. The stiffening/softening of the combined fluid mass-pylon system is manifested as a function of fluid decrease/increase in the container. Then, the time histories at any station on the pylon for ground-induced motions can be computed by modal analysis. Finally, if structural identification is the goal, then the aforementioned time histories can be used as raw date, followed by application of the FT with Hanning windows sequentially places across the time axis.

In general, the methodology developed herein is applicable to other types of structural systems with transient mass attachments and to general types of dynamic loads, including the case of a moving mass. A common case is the fixed, lumped mass at the top, which may also act as a secondary system in the case of light-weight appendages. It is also possible to augment the present formulation to model a compliant foundation by introducing equivalent soil springs at the base of the pylon, and finally to treat beams with a non-uniform cross-section. One field of application of this work is SHM for pylons used in electrified rail lines, where high frequency ground vibrations are induced in the vertical direction due to the passage of fast trains. In this case the presence of a time-variable lumped mass at the top of the pylon may either be detrimental or beneficial, depending on the frequency content of the external excitation and on the rate by which this mass increases or decreases.

**Author Contributions:** Conceptualization, G.D.M. and G.I.D.; methodology, G.D.M. and G.I.D.; software, G.D.M. and G.I.D.; validation, G.D.M. and G.I.D.; formal analysis, G.D.M. and G.I.D.; investigation, G.D.M. and G.I.D.; resources, G.D.M. and G.I.D.; writing—original draft preparation, G.D.M. and G.I.D.; writing—review and editing, G.D.M. and G.I.D.; visualization, G.D.M. and G.I.D.; supervision, G.D.M. All authors have read and agreed to the published version of the manuscript.

**Funding:** The authors acknowledge support by the German Research Foundation (DFG) through grant SM 281/20-1 and the Hellenic Foundation for Research and Innovation (HFRI) under the third call for PhD fellowships (Fellowship Number: 6522). Any opinions, findings, conclusions, or recommendations expressed in this paper are those of the authors and do not necessarily reflect the views of the DFG or the HRFI.

**Data Availability Statement:** Data is contained within the article.

**Conflicts of Interest:** The authors declare no conflict of interest.

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
