# Peer review of "On the Numerical Treatment of the Temporal Discontinuity Arising from a Time-Varying Point Mass Attachment on a Waveguide"

_algorithms, doi:10.3390/a16010026_

Round 1

Reviewer 1 Report

    Dear Editor, 

    "On the numerical Treatment of the Temporal Discontinuity Arising from a Time-varying Point Mass Attachment on a Waveguide" is the topic of the authors' work. They examined two eigenvalue extraction methods namely the Laguerre and the QR algorithms. The aim of the analysis is to identify the most efficient technique for interpreting time signals registered at a given pylon as a means for detecting damage, a procedure that finds application in structural health monitoring of civil engineering infrastructure.

    The structure of the manuscript is clear and the analysis is acceptable. In addition to minor corrections:

  ∙  Page 1, line 9: change "modelled" by "modeled";

  ∙  Page 1, line 19: change " registered at given pylon" by " registered at a given pylon";

  ∙  Page 1, line 29: change "with time in a scale comparable" by "with time on a scale comparable";

  ∙  Page 1, line 32: change "modelled as waveguides" by "modeled as waveguides";

  ∙  Page 1, line 39: change "a vibro-impact nonlinear" by "a Vibro-impact nonlinear";

  ∙  Page 1, line 40: change "new methods of solution must be sought," by "new methods of the solution must be sought,";

  ∙  Page 1, line 40: change "new methods of solution must be sought" by "tnew methods of solving the problem must be sought";

  ∙  Page 1, line 40: change "As consequence" by "As a consequence";

  ∙  Page 1, line 42: change "For heavy attached masses" by "For heavily attached masses".

  ∙  Page 1, line 43: change "In here, we solve for" by "Here, we solve for";

  ∙  Page 2, line 51: change "from solution of single degree-of-freedom (SDOF) equations" by "the solution of single-degree-of-freedom (SDOF) equations";

  ∙  Page 2, line 53: change "are considered, because the motion" by "are considered because the motion";

  ∙  Page 2, line 55: change "at the same time instant" by "at the same instant";

  ∙  Page 2, line 57: change "then the new eigenproperties of " by "then the new eigen properties of ";

  ∙  Page 2, line 62: change "and torsional response" by "and torsional responses";

  ∙  Page 2, line 76: change "is sizeable as compared" by " is large as";

  ∙  Page 2, line 77: change "change in reference to the original" by "change compared to the original";

  ∙  Page 2, line 84: change "Focusing on the eigenvalue analysis the combined pylon-mass system," by "Focusing on the eigenvalue analysis of the combined pylon-mass system";

  ∙  Page 3, line 114: change "In reference to the dynamic" by "In terms of the dynamic";

  ∙  Page 3, line 116: change "differential operator or to the boundary conditions" by "differential operator or the boundary conditions";

  ∙  Page 3, line 119: change "the structural system is labelled as non-autonomous" by "the structural system is labeled as non-autonomous";

  ∙  Page 3, line 122: change "we focus on longitudinal vibrations" by "we focus on the longitudinal vibrations";

  ∙  Page 4, line 154: change "with a time varying mass" by "with a time-varying mass";

  ∙  Page 4, line 168: change " for the pylon modelled " by "for the pylon modeled";

  ∙  Page 5, line 176: change " Note the summation " by " Note that the summation";

  ∙  Page 5, line 183: change "In the above, [M(t)], [C], [K] are the mass matrix" by "In the above, [M(t)], [C] and [K] are the mass matrix";

  ∙  Page 5, line 188: change " as a first order differential equation" by " as a first-order differential equation";

  ∙  Page 5, line 190: change "At this point we opt for a time-stepping solution" by "At this point, we opted for a time-stepping solution";

  ∙  Page 5, line 193: change "system matrix" by "the system matrix".

  ∙  Page 8, line 255: change "the Laguerrre algorithm" by "the Laguerre algorithm";

  ∙  Page 10, line 269: change " is solved for every time step" by " is solved every time step";

  ∙  Page 10, line 272: change "and the right column for the increasing mass rate" by "and the right column is for the increasing mass rate";

  ∙  Page 17, line 386: change "application of the short time" by "the application of the short-time";

  ∙  Page 19 , References [25,26,27] should be revised.

Reviewer 2 Report

In the reviewed manuscript, oscillations of a cantilevered pylon with attached time-varying mass are treated numerically. For this purpose, a numerical approach is proposed by the authors and illustrated by certain numerical results. The paper content is within the scope of Algorithms journal and might of potential interest to its audience. However, certain questions arise regarding the paper content:

1) Could the authors, please, specify more clearly what is the particular novelty of their manuscript? The information provided in Section 3 might be treated as review of existing techniques (i.e., already realized in LAPACK numerical library), the application of STFT is already a classical numerical technique. The problem statement itself is also questionable especially regarding the numerical example provided - with such a big attached mass and its fast growth/decrease, this process definitely induce some non-linear effects - this question should be carefully discussed.

2) In the conclusion section it is stated that "This work investigates the  accuracy, efficiency and best applicability ...". However, to the reviewer's opinion, nothing of this could be found in previous sections. Could the authors provide more details regarding this statement?

3) There is a lack of any verification/comparison with respect to independent numerical results / results provided by other numerical techniques / experimental data. Without such comparison, the validity of the obtained results is questionable. 

Round 2

Reviewer 2 Report

The authors have carefully responded to almost all the comments. However, unfortunately, the question regarding the numerical/experimental verification of the developed approach over the independent data or alternative computational technique (probably, some less complex problem could be considered for comparison) though being briefly responded in the "Cover letter" is not considered in the revised version of the manuscript.

Therefore, it is strongly suggested to the authors to provide any validation/verification comparisons in the current paper. That would sufficiently strengthen it.
